

# Deriving the slit functions from OMI solar observations and its implications for ozone-profile retrieval

Kang Sun[1], Xiong Liu[1], Guanyu Huang[1], Gonzalo González Abad[1], Zhaonan Cai[1], Kelly Chance[1], and Kai Yang[2]

[1]Harvard-Smithsonian Center for Astrophysics, 60 Garden Street, Cambridge, MA, USA
[2]Department of Atmospheric and Oceanic Science, University of Maryland, College Park, MD, USA

*Correspondence to:* Kang Sun (kang.sun@cfa.harvard.edu)

**Abstract.** The Ozone Monitoring Instrument (OMI) has been successfully measuring the Earth's atmospheric composition since 2004, but the on-orbit behavior of its slit functions has not been thoroughly characterized. Preflight measurements of slit functions have been used as a static input in many OMI retrieval algorithms. This study derives on-orbit slit functions from the OMI irradiance spectra assuming various function forms, including standard and super Gaussian functions and a stretch to the preflight slit functions. The on-orbit slit functions in the UV bands show U-shaped cross-track dependences that cannot be fully represented by the preflight ones. The full widths at half maximum (FWHM) of the stretched preflight slit functions for detector pixels at large viewing angles are up to 20 % larger than the nadir pixels for the UV1 band and 5 % larger for the UV2 band. Nonetheless, the on-orbit changes of OMI slit functions are found to be insignificant over time after accounting for the solar activity, despite of the decaying of detectors and the occurrence of OMI row anomaly. Applying the derived on-orbit slit functions to ozone-profile retrieval shows substantial improvements over the preflight slit functions based on comparisons with ozonesonde validations.

## 1 Introduction

The Dutch-Finnish Ozone Monitoring Instrument (OMI) on board the NASA Aura satellite has been measuring the direct sunlight and backscattered sunlight from the Earth since 2004. Spectrally, the OMI instrument incorporates two 2-D charge-coupled device (CCD) detectors for the ultraviolet (UV) and visible (VIS) bands. The UV band is optically separated into the UV1 band (264–311 nm, $\sim 0.6$ nm resolution) and the UV2 band (307–383 nm, $\sim 0.4$ nm resolution), and the VIS band covers 349–504 nm at $\sim 0.6$ nm resolution (Dirksen et al., 2006). Spatially, the OMI instrument has a wide, $115\,^\circ$ field of view, which is divided into 30 cross-track positions in the UV1 band and 60 cross-track positions in the UV2 and VIS bands. At the nadir point, the spatial resolution is $13 \times 48$ km$^2$ for UV1 and $13 \times 24$ km$^2$ for UV2/VIS, significantly finer than the nadir resolutions of existing spaceborne measurements over similar spectral ranges like GOME, SCIAMACHY, GOME-2, and OMPS. The spectral and spatial coverages of OMI enable retrievals of various key constituents of the Earth's atmosphere, including ozone, nitrogen dioxide, sulfur dioxide, formaldehyde, BrO, water vapor, and many others, with daily global coverage at the equator (Levelt et al., 2006). In addition to abundances of atmospheric species, OMI delivers products like cloud



fraction/height, aerosol optical depth, surface UV radiation, and solar activity proxies (Tanskanen et al., 2006; Torres et al., 2007; Vasilkov et al., 2008; DeLand and Marchenko, 2013).

The slit function, also called the instrument transfer function (ITF), instrumental spectral response function (ISRF), or instrument line shape (ILS) in the literature, is the instrument's response to a Dirac delta function in the spectral domain (Dirksen

et al., 2006; Beirle et al., 2017; Sun et al., 2017). Therefore, the OMI observed spectra can be modeled as a convolution of the OMI slit functions and the native resolution spectra. A thorough understanding of the OMI slit function is crucial to accurately modeling the OMI spectra and retrieving atmospheric constituents/properties. The OMI slit functions vary in both the spectral dimension (columns) and the spatial dimension (rows, or cross-track direction) of the 2-D detectors and have been measured accurately in a preflight experiment (Dirksen et al., 2006). These preflight slit functions have been adopted in a wide range of

operational OMI retrieval algorithms (Kurosu et al., 2004; Veefkind et al., 2006; Chan Miller et al., 2014; Wang et al., 2014; González Abad et al., 2015; van Geffen et al., 2015; Li et al., 2017). However, it is still unclear if the preflight slit functions adequately represent the on-orbit slit functions for OMI, given the impact of the launching process and the contrasts between laboratory and space conditions. Additionally, the on-orbit slit functions may evolve over time due to instrumental issues, as observed in some other satellite instruments (De Smedt et al., 2012; Miles et al., 2015; Beirle et al., 2017; Sun et al., 2017).

The optical degradation of OMI has been markedly small over the mission, but the row anomaly (RA) appeared in 2007 and had made significant impact on about one third of cross-track positions since January 2009 (Schenkeveld et al., 2017). The impact of RA on the OMI slit functions is still poorly understood.

The SAO ozone-profile retrieval algorithm derives on-orbit slit functions from averaged OMI solar observations in 2005–2007, assuming standard Gaussian slit function form, which showed better performance than using the complex preflight slit

functions (Liu et al., 2010). This indicates that although no significant changes in the OMI slit functions have been noted over the mission, the on-orbit slit functions may differ from the preflight at the ozone retrieval windows. Therefore, it is necessary to reevaluate the OMI slit functions using on-orbit data. To this end, we investigate the temporal variation of the slit function using more than 10 years of OMI irradiance data, compare the on-orbit slit functions derived from irradiance with the preflight ones, and evaluate multiple on-orbit slit function forms by validating the ozone profile retrieved using different slit functions

with ozonesonde observations.

## 2  Instrument and data analyses

### 2.1  OMI instrument and its solar measurements

The OMI instrument is a push-broom grating spectrometer flying in a 705-km sun-synchronous polar orbit with ~13:45 equator-crossing time. The two 2-D CCD detectors have 780 columns (the spectral dimension, along the flight direction)

and 576 rows (the spatial dimension, perpendicular to the flight direction). In the global mode that is mostly used in Earth observation, the 576 spatial pixels are selectively binned into 60 (UV2/VIS) or 30 (UV1) rows, corresponding to the same numbers of cross-track positions at the Earth's surface. Anomalous behavior of some rows started in June 2007 and became much more significant in January 2009. This so-called "row anomaly" (RA) permanently affects radiances of rows 25–42 (out




of 1–60) and 54–55 of the UV2/VIS bands and occasionally affects rows 43–53. The RA rows in UV1 have similar relative positions, although all UV1 rows are affected in the northern parts of orbits. The RA is believed to be caused by loose thermal insulating materials partially blocking OMI's Earth-observing field of view. Detailed description of the RA can be found at Schenkeveld et al. (2017).

OMI also has an irradiance view port for solar calibration. Solar spectra, shown in Fig. 1, are measured once per day near the northern terminator of an orbit. The direct sunlight is attenuated by an optical mesh, then reflected by one of the solar diffusers (quartz volume diffuser, regular aluminum, or backup aluminum), and finally reflected by a folding mirror to the remainder of the optical system that is identical to the Earth radiance measurement. The blocking effects that caused the RA in the radiance measurement are not observed in the irradiance; the only noted RA-related impact on irradiance is that the RA rows in the UV1
band show faster optical degradation due to additional solar exposure from RA reflections (Schenkeveld et al., 2017).

The OMI solar irradiance is also used to monitor the solar activity through the variations of deep solar lines, which generally get shallower when the Sun is more active. The most used solar activity proxies are the core-to-wing indices of the Mg II line at 280 nm, Ca II K line at 393.4 nm, and Ca II H line at 396.8 nm (labeled in Fig. 1). The OMI Mg II index varied by $\sim 10\%$ between solar cycle minimum and maximum. Although the Ca II indices are well correlated with the Mg II index, their relative
variations are smaller by factors of 7–9 (DeLand and Marchenko, 2013). The irradiance spectrum at other wavelengths has weaker correspondence with the solar cycle, generally $< 0.2\%$ between cycle minimum and maximum in the UV2/VIS bands and 0.5–1% in the UV1 band (Marchenko et al., 2016).

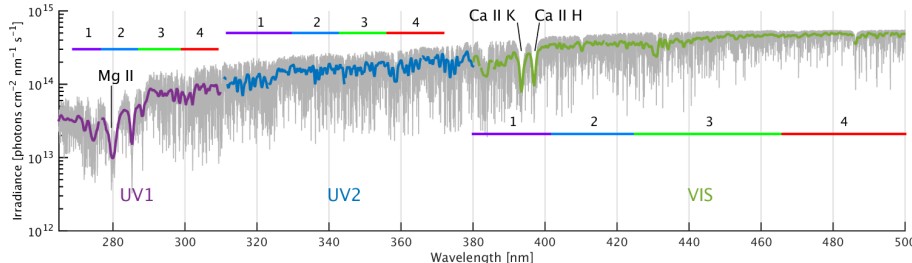

**Figure 1.** Solar irradiance spectra observed by the UV1, UV2, and VIS bands of OMI (overlapped spectral regions are not shown). The background high-resolution solar reference spectrum in gray color is from Dobber et al. (2008). Deep solar lines used to monitor solar activity are labeled. The colored horizontal bars indicate the spectral windows within which slit functions are fitted in Sect. 4.

### 2.1.1    OMI slit functions

The OMI slit functions were determined during the preflight characterization for each spectral pixel and cross-track position.
The measured slit functions at discrete wavelengths were then fitted using a combination of a standard Gaussian and a flat-top function, defined at a wavelength grid $\Delta\lambda$:

$$S_{\mathrm{pre}}(\Delta\lambda) = A_0 \exp\left[-\left(\frac{\Delta\lambda - \lambda_0}{w_0}\right)^2\right] + A_1 \exp\left[-\left(\frac{\Delta\lambda - \lambda_1}{w_1}\right)^4\right], \tag{1}$$





where $A_0/A_1$ is the relative amplitude of the standard and flat-top Gaussian components; $\lambda_0$ and $\lambda_1$ are their central positions; and $w_0$ and $w_1$ determine their widths. For the UV1 band, only the standard Gaussian component is necessary ($\lambda_0$ and $A_1$ are zero). The accuracies of these preflight slit functions were demonstrated to be better than $2\%$ within $\pm 2$ FWHM (full width at half maximum) and $\sim 3\%$ between $\pm 2$ FWHM and $\pm 3$ FWHM during the preflight test (Dobber et al., 2008).

Information on the on-orbit slit function can be retrieved by fitting the observed solar irradiance with a high-resolution solar reference spectrum (shown in Fig. 1) and some assumed slit function forms. Wavelength shift/squeeze terms and a polynomial baseline are also included in the fitting. Two high-resolution solar reference spectra are tested: the KNMI spectrum (0.025 nm resolution and 0.01 nm sampling, Dobber et al., 2008) and the SAO2010 spectrum (0.04 nm resolution and 0.01 nm sampling, Chance and Kurucz, 2010). They give very similar results for the derived slit functions. The KNMI spectrum is used

in the following results due to its better radiometric calibration in the OMI spectral range. This slit function fitting method has been described in Sun et al. (2017), where multiple analytical and numerical function forms were tested to represent the slit functions of the OCO-2 instrument. These function forms were also tested for OMI. The (a)symmetric standard Gaussian, super Gaussian, and fitting a homogeneous stretch to the preflight slit functions were found to produce stable fitting results. The other function forms (stretch/sharpen, hybrid Gaussians) became increasingly unstable later in the OMI mission, when the

signal-to-noise ratio (SNR) of solar spectra was degraded. The Gaussian function family can be generalized as the asymmetric super Gaussian function (Beirle et al., 2017):

$$S(\Delta\lambda) = \exp\left(-\left|\frac{\Delta\lambda}{w + \mathrm{sgn}(\Delta\lambda)a_w}\right|^k\right), \tag{2}$$

where $w$ is the half width at $1/e$, $k$ is the shape parameter, $a_w$ is the asymmetry parameter, and $\mathrm{sgn}()$ is the sign function. If $k$ is fixed at 2, the slit function is standard Gaussian; if $a_w \neq 0$, the slit function is asymmetric. The homogeneously stretched

preflight slit function is simply

$$S(\Delta\lambda) = S_{\mathrm{pre}}\left(\Delta\lambda/\mathrm{stretch}\right). \tag{3}$$

Examples of the preflight slit functions and the fitted function forms are illustrated in Fig. 2 for the fitting window 343–356 nm in the UV2 band.

Information on the slit function fitting algorithm, the slit function is generally assumed to be constant within the fitting spectral window.

When fitting a stretch to the preflight slit function, the preflight slit function at the median wavelength is applied to the entire window. Spectrally resolved stretch of the preflight slit functions was also tested by stretching spectrally dependent preflight slit functions over sliding and overlapping windows, but the results were influenced by radiometric and solar reference spectrum uncertainties over short wavelength ranges. The ozone profiles retrieved assuming spectrally constant slit functions also outperform those retrieved using spectrally resolved slit functions. Consequently, the derived slit functions are assumed to be

spectrally constant over each fitting window.



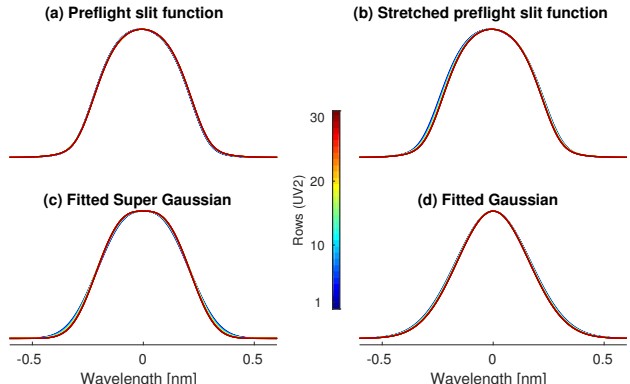

**Figure 2.** (a) Preflight slit functions at the median wavelength of the spectral window 343–356 nm (349.5 nm). The spectral variations of preflight slit functions over this window are small (see the second row of Fig. 5 for the range of preflight slit function width in UV2). (b-d) Derived slit functions for the spectral window 343–356 nm by fitting a stretch factor to the preflight slit function, super Gaussian function, and standard Gaussian function, respectively. The solar spectrum used in the fitting is the average OMI irradiance from 2005 to 2007. Adding the asymmetric parameter in the super/standard Gaussian fitting gives little difference. Only half of the UV2 cross track positions (1–30) are shown as the other half are essentially a mirror image (see the second row of Fig. 5).

## 2.2 SAO OMI ozone-profile retrieval algorithm and validation

The optical path of the solar irradiance and the earth radiance measurements are very similar, but it is still an open question whether the slit functions are identical for the solar and earthshine spectra. Besides, the derived on-orbit slit functions are sometimes simplified compared to the preflight ones. Therefore, it is necessary to apply the on-orbit slit functions derived from solar irradiance to the earthshine spectra fitting algorithm and validate the retrieval results. In this study, the SAO OMI ozone-profile retrieval algorithm is used to test different options of slit functions. The OMI ozone-profile retrievals have substantially higher relative accuracy than other OMI products (e.g., $SO_2$, $NO_2$, HCHO, etc.), and there have been extensive validations for the ozone-profile product. Therefore, the OMI ozone-profile retrieval is the first choice to test the on-orbit slit functions.

The SAO OMI ozone-profile retrieval algorithm was described in detail by Liu et al. (2010) with further improvements by Bak et al. (2013, 2016) and validated extensively against ozonesonde and MLS observations (Huang et al., 2017a, b). In the operational SAO ozone-profile algorithm, partial ozone columns (in Dobson Unit, DU, 1 DU = $2.69 \times 10^{16}$ molecules $cm^{-2}$) are retrieved at 24 layers from the surface to about 60 km using the optimal estimation technique. Stratospheric ozone column (SOC) and tropospheric ozone column (TOC) are derived from the OMI ozone profile using the thermal tropopause heights, defined by the lapse rate criterion (WMO, 1957), from the National Center for Environmental Protection (NCEP) reanalysis. The total degree of freedom for signal (DFS) of the retrieved ozone profile is 6–7 with 5–7 in the stratosphere and 0–1.5 in the troposphere. Fitting windows in both UV1 (270–309 nm) and UV2 (311–330 nm) are used to retrieve ozone abundance at different altitudes. Wavelengths around Mg II (280 nm) and Mg I (285 nm) lines are not included in the retrieval. Because of the mismatch of cross-track positions between UV1 and UV2, UV2 spectra at every two adjacent cross-track positions are





co-added to match the UV1 spatial resolution. When applying the preflight slit functions, they are also averaged every two cross-track positions for UV2. Hence the retrievals are performed at the UV1 spatial resolution. In addition, OMI radiances are pre-calibrated based on two-day-average radiance differences in the tropics between OMI spectra and spectral simulations using zonal mean MLS ozone profiles at pressure less than the 215 hPa level and climatological ozone profiles at pressure

greater than the 215 hPa level. This "soft calibration" significantly reduces the OMI L1B calibration errors that are dependent on both wavelength and cross-track positions.

   In this study, the ozone profiles are retrieved using four different options of slit functions: the preflight as well as standard Gaussian, super Gaussian, and stretched preflight derived from OMI irradiance. The soft calibration is turned off to make fair comparisons between slit functions, because the soft calibration algorithm is currently only implemented with standard

Gaussian slit functions. The other retrieval options are kept the same as the operational algorithm whenever it is possible. This slit function comparison is named as the "two-band" case. Another case is also tested by using only the UV2 window (311–330 nm). In this "UV2-only" case, there are no compounding factors induced by averaging the adjacent UV2 spectra and preflight slit functions. The OMI ozone profiles are retrieved at 60 cross-track positions, instead of 30 cross-track positions for the two-band case. A "UV1-only" case is also tested with a subset of ozonesondes, but found to be insensitive to different

options of slit functions (although the on-orbit/preflight slit functions are quite different for the UV1 band, see the first row of Fig. 5).

   Ozonesonde observations are widely used to validate satellite ozone-profile retrievals (Jiang et al., 2007; Worden et al., 2007; Froidevaux et al., 2008; Kroon et al., 2011; Wang et al., 2011; Jia et al., 2015; Huang et al., 2017a). We use the same global ozonesonde dataset described by Huang et al. (2017a) but only during 2004–2008. The ozonesonde profiles extend

from the surface up to $\sim$ 35 km with vertical resolution of 100–150 m, 3–5% precision, and 5–10% accuracy (Komhyr, 1986; Komhyr et al., 1995; Johnson et al., 2002; Smit et al., 2007; Deshler et al., 2008). When compared to the OMI ozone profiles that have much lower vertical resolution, ozonesonde profiles are first integrated into the corresponding OMI layers and then degraded to the OMI vertical resolution using the OMI a priori ozone profile and Averaging Kernels (AK). The locations of the ozonesondes are shown in Fig. 3, where the size of circles denotes the number of successful ozonesonde validations for the

two-band case in 2004–2008, and the color denotes mean bias between the tropospheric ozone column retrieved in the two-band case using standard Gaussian slit functions and the tropospheric ozone column from ozonesonde with OMI AK applied. A successful validation is defined when OMI retrievals using different slit functions and the collocated ozonesonde profile all pass the filtering criteria. The OMI/ozonesonde data filtering criteria in this study are very similar to Huang et al. (2017a): OMI effective cloud fraction less than 0.3; OMI solar zenith angle less than 75 °; ozonesondes that reach an altitude of at least 30

km and have data gaps no greater than 3 km; ozonesonde correction factors (CFs), if exist, in the range of 0.85 to 1.15. We did not apply these CFs because it is not clear that they should be applied to the ozone profiles, especially for the troposphere, and CFs are only available for a limited fraction of ozonesondes (Morris et al., 2013). For each filtered ozonesonde profile, the nearest filtered OMI profile within ±1 ° latitude, ±1 ° longitude, and ±6 hours is used for validation on the individual profile basis. The validation in this work also has the following important differences from Huang et al. (2017a):



(1) Huang et al. (2017a) used the operational ozone-profile product that coadded four spatial pixels along the track and hence had a nadir spatial resolution of $52 \times 48$ km$^2$; this study retrieves ozone profiles only at pixels collocated with ozonesonde stations with standard spatial resolution of the UV1 band ($13 \times 48$ km$^2$ at nadir) for the two-band case and resolution of the UV2 band ($13 \times 24$ km$^2$ at nadir) for the UV2-only case. As the soft calibration is turned off, the overall biases are slightly larger than the operational SAO ozone profile product.

(2) As will be shown in Sect. 3–4, the cross-track discrepancies between on-orbit and preflight slit functions are much more significant than the temporal variations of on-orbit slit functions over the OMI mission. Hence only the "pre-RA" period (2004–2008) is included in the validation to consistently compare all cross-track positions.

(3) Huang et al. (2017a) only used cross-track positions 4–27 at the UV1 spatial resolution, whereas in this study all cross-track positions are included in the validation. The successful validations are further grouped according to the cross-track positions of the OMI ozone profiles. Figure 4 shows the number of successful validations in the two-band case and UV2-only case over 2004–2008. On average, there are 135 successful validations per UV1 cross-track position for the two-band case and 74 successful validations per UV2 cross-track position for the UV2-only case.

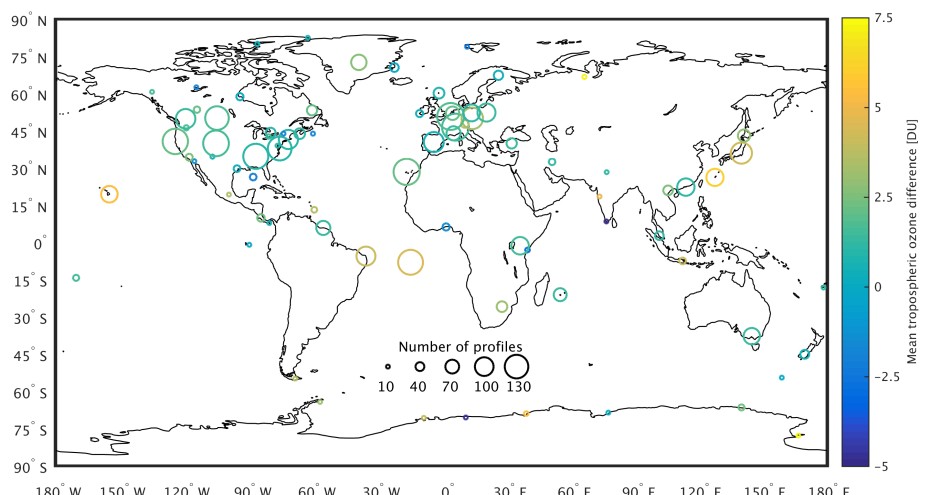

**Figure 3.** The locations of the ozonesondes used in this study. The size of circles denotes the number of successful validations for the two-band case in 2004–2008, and the color denotes mean bias between the tropospheric ozone column from OMI and the ozonesonde convolved with the OMI AK.

## 3 Difference between on-orbit and preflight slit functions

The OMI solar irradiance is generally assumed to be stable, and many retrieval algorithms use the average or the first principle component of OMI irradiance spectra over multiple years to enhance the SNR (Liu et al., 2010; Wang et al., 2014; González Abad et al., 2015). Figure 5 compares the FWHM of preflight slit functions with those of on-orbit slit functions derived from the average OMI irradiance over 2005–2007 at all three OMI bands. The ranges of preflight slit function FWHM





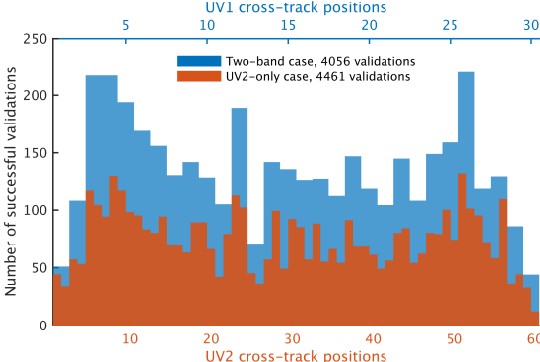

**Figure 4.** Number of successful validations at different cross-track positions for the two-band case (blue) and UV2-only case (red) over 2004–2008.

within each fitting window are denoted by gray areas. Adding asymmetry factors in the on-orbit slit function fitting makes little difference in all fitting windows, indicating that the OMI slit functions are sufficiently symmetric, as noted by Beirle et al. (2017). For the UV1 band (the first row of Fig. 5), the preflight slit functions are Gaussian, so only the standard Gaussian functions forms are fitted (fitting a stretch to the preflight slit functions is identical to fitting a standard Gaussian). At the low

wavelength end (270–277 nm), the on-orbit slit functions are remarkably broader than the preflight ones toward the edges of cross-track positions, by up to $20\%$. Similar effects are observed at the high wavelength end of the UV1 band (299–309 nm), although much less significant.

Panels in the second row of Fig. 5 show the results for the UV2 band, divided into four fitting windows, with the super Gaussian included. The on-orbit and preflight slit functions at the first half of the cross-track positions (1–30) for window 3 are

plotted in Fig. 2. The cross-track patterns are similar for all four fitting windows. The preflight slit functions show very little cross-track variation ($< 1\%$ in FWHM), whereas the FWHMs of standard Gaussians and the stretched preflight slit functions show a U-shaped cross-track dependency. The derived slit function FWHM at large off-track viewing angles are up to $5\%$ broader than the nadir ones. The FWHM of super Gaussian is only weakly cross-track dependent, but the shape parameter $k$ has a strong reverse-U-shaped cross-track dependency. As illustrated more clearly in Fig. 2, the derived on-orbit slit functions

are broader toward the larger off-track viewing angles if only the widths are fitted; when the slit function shape can also be adjusted in the super Gaussian fitting, the broadening is redistributed towards the tails of the slit functions.

Panels in the third row of Fig. 5 show the results for the VIS band. The preflight slit functions also have very little cross-track variation; the on-orbit slit functions capture some cross-track dependency but not as significant as the UV2 band. Fitting window 1 is an exception, where a reverse-U-shaped stretch factor and a U-shaped shape parameter $k$ are present.




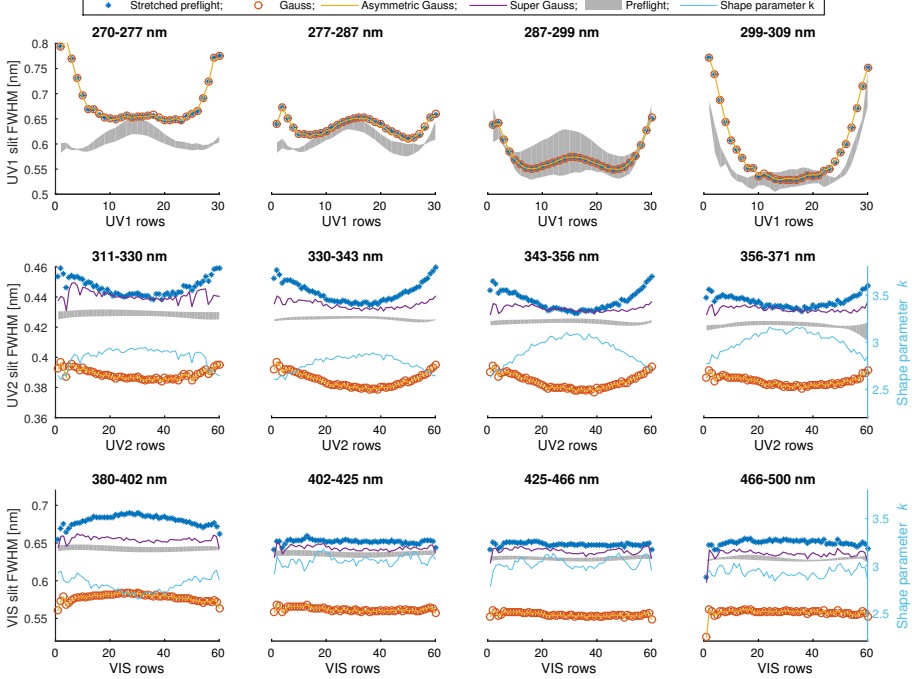

**Figure 5.** Comparisons of the FWHMs of on-orbit slit functions derived from the average OMI irradiance over 2005–2007 with the preflight ones for all three OMI bands, each divided into four fitting windows as shown in Fig. 1. Because the preflight slit functions vary continuously with wavelength, the ranges of preflight slit function FWHMs within the spectral window are plotted as the gray bands. The super Gaussian shape parameter $k$ for the UV2 and VIS fitting windows are also shown with vertical axises on the right.

## 4   Temporal variation of OMI slit functions

The on-orbit variations of OMI slit functions have widely been deemed insignificant throughout the mission. By deriving on-orbit slit functions using the daily OMI solar irradiance, it is possible to verify this assumption. Figure 6 presents the evolution of derived slit function widths for all cross-track positions from September 2004 to May 2016, where the three OMI bands are each divided into four fitting windows (see Fig. 1 for the window locations), and a standard Gaussian slit function is derived for each window. The super Gaussian and stretched preflight fits are also tested at selective cross-track positions and fitting windows. The fitted stretch factor and shape factor $k$ of super Gaussian have essentially the same temporal trends as the width of standard Gaussian, so only the widths of derived standard Gaussian are used to represent the slit function change. Significant temporal variations of the slit function widths can be observed at windows 1–2 of the UV1 band (270–287 nm, rows 1–2, column 1 of Fig. 6) and window 1 of the VIS band (380–402 nm, row 1, column 3 of Fig. 6).

We assume that the high-resolution solar reference spectrum stays constant, because there have been no high-resolution solar reference spectra that incorporate the solar cycles available. However, this assumption does not hold near deep solar lines and at short wavelengths in the UV at the OMI resolution (Marchenko and DeLand, 2014; Marchenko et al., 2016). Since the




true solar irradiance is assumed to be invariant, the relative changes of observed solar lines following the solar cycles will be interpreted as the slit function change in the fitting. As a result, the temporal variations seen in Fig. 6 may result from solar cycles and not necessarily indicate the real changes of on-orbit slit functions.

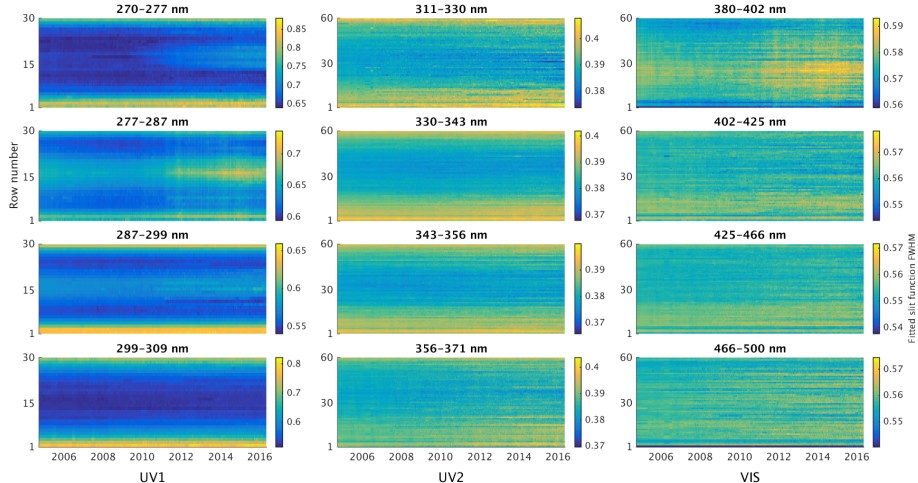

**Figure 6.** Temporal variations of derived standard Gaussian slit function FWHM for all cross-track positions from September 2004 to May 2016. Three OMI bands (columns of the plot) are each divided into four fitting windows (rows of the plot). Spectral coverages of each window are also labeled in the title of each subplot.

The solar activity can be represented by the OMI Mg II core-to-wing index, defined as the ratio between OMI irradiance at 280 nm and the average irradiances at 277 and 283 nm. The OMI Mg II index has also been verified against other ground-based and space-borne solar observations (DeLand and Marchenko, 2013). The Mg II index is modulated by a distinct 11-year solar cycle and a 27-day solar rotation cycle, whose amplitude also varies due to variations in sunspots. Given the complex nature of solar activity, it is highly unlikely that any potential factors that may contribute to slit function change (e.g., instrument degradation, RA effects) are correlated with the Mg II index. To separate the spurious slit function changes due to insufficient consideration of solar activity with potential real changes, the derived slit function widths are plotted against the Mg II index for the aggregated rows that are not influenced by RA (non-RA rows) and influenced by RA (RA rows) in Figs. 7-8, color-coded by time over 2004–2016. For the non-RA rows, the derived on-orbit slit function widths show linear correlations with the Mg II index for most windows throughout the mission (Fig. 7). Window 4 at UV1 is the only exception, where the derived width abruptly increased by ∼ 7 % in April 2008. This change cannot be explained by solar activity or major RA events but occurred exactly when the SNR of the UV1 band started to decrease (Huang et al., 2017a), so it is still possibly a secondary RA effect. The slopes between the relative changes of slit functions widths and relative change of the Mg II index are also labeled in Fig. 7. The slope is the largest (0.50–0.51, 95 % confidence interval) for window 2 of the UV1 band (where the Mg II line is located), meaning that as the Mg II index varied by 10 % from solar cycle minimum to maximum, the derived slit function width varied by 5 %. It is followed by window 1 of the UV1 band (0.24–0.25) and window 1 of the VIS band (0.13). Window




1 at UV1 has the shortest wavelength, and window 1 at VIS contains the strong Ca solar lines, so these two windows are also strongly influenced by solar activity. Most of the derived slit function changes can be explained by the Mg II index and hence solar activity variations (see the $R^2$ in the plot), and the residual variations are mostly random noise that increases with time as the instrument slowly degrades. Therefore, the apparent temporal variations in Fig. 6 are due to solar activity, instead of real

5  on-orbit slit function changes.

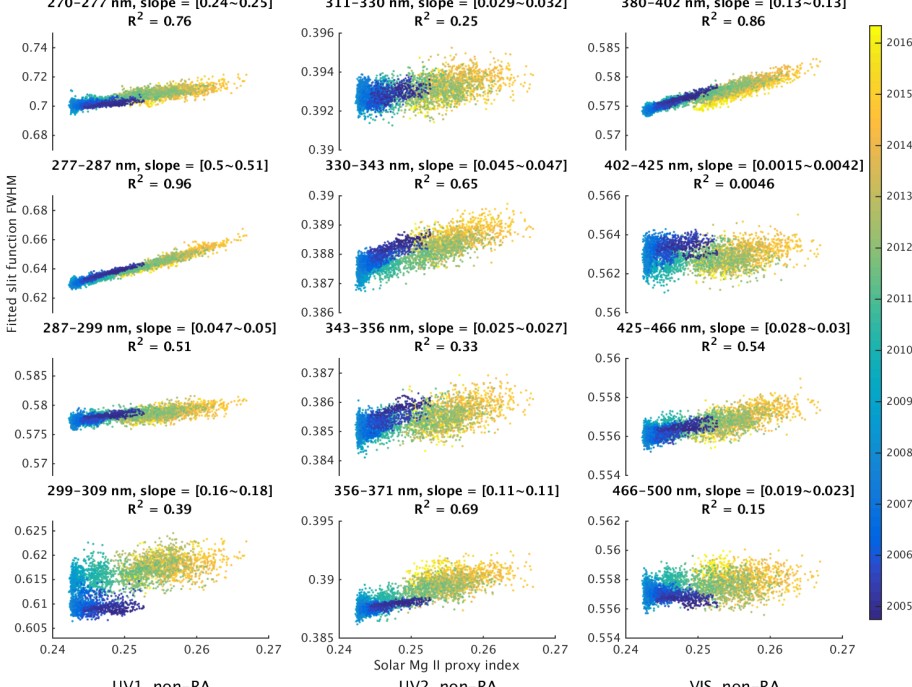

**Figure 7.** Scatter plots between the average slit function FWHM of all non-RA rows (assuming standard Gaussian) and the Mg II index. The columns and rows are the same as Fig. 6. The linear fitting slopes between the normalized slit function widths and the normalized Mg II index are shown in the title of each subplot.

Although no significant RA impact on the derived slit functions can be observed from Fig. 6, the impact is visible when plotting the slit function widths of the RA rows against the Mg II index (Fig. 8, in the same format as Fig. 7). Coincidently, both the solar minimum (also the minimum of the Mg II index) and the major RA events occurred in 2008–2009. Hence the impact of RA will be manifested as hysteresis in the scatter plots. In contrast to the non-RA rows, the derived slit function

10  widths of the RA rows show different relationship with the Mg II index before and after the major RA event in January 2009, mostly notable at windows 1–3 of the UV1 band. Some extra hysteresis is observable but small (0.3 % of slit function width) at window 2 of the UV2 band and window 2 of the VIS band. Overall, the temporal variations of slit function widths for non-RA rows, if they exist, are no more than 1 % and much smaller than differences between the preflight and on-orbit slit functions for





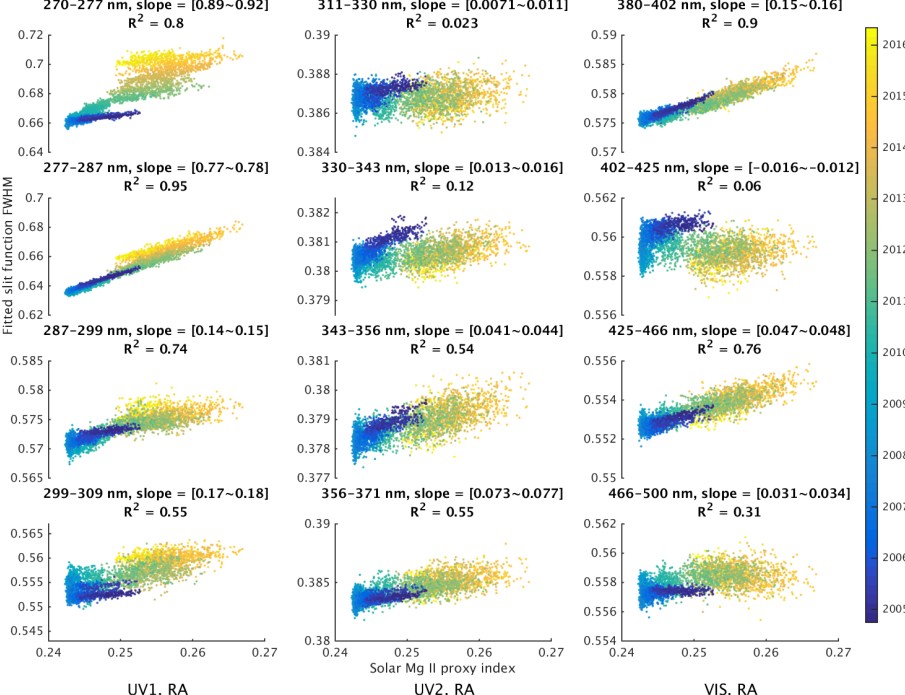

**Figure 8.** The same as Fig. 7 but for the RA rows. The ranges of vertical axises are the same as Fig. 7 as well for easier comparison.

all three OMI bands (Fig. 5). The RA rows also did not show any significant temporal variations before the major RA events in 2009.

## 5 Validation of ozone profiles retrieved using different slit functions

### 5.1 The two-band case

5     Figure 9 compares the SOC (first row), TOC (second row), and the tropospheric ozone column from surface to 500 hPa (TOC500, third row) of OMI retrieved using four different options of slit functions with the ozonesonde data over 2004–2008. The standard/super Gaussian and the stretch preflight slit functions are derived using the mean solar irradiance over 2005–2007, and the slit functions are assumed to be invariant over time. For the SOC, the correlation coefficients between OMI and ozondsondes are similar for all slit functions, whereas the stretched preflight and super Gaussian options show smaller mean

10    absolute biases ($-0.11$ and $-0.19$ DU) than the standard Gaussian option ($0.60$ DU), relative to the ozonesonde data. The preflight option shows the largest mean absolute bias ($-2.01$ DU) and the largest standard deviation ($13.48$ DU). For the TOC, the standard Gaussian retrieval shows the lowest mean bias ($1.89$ DU), followed by stretched preflight ($2.59$ DU) and the super Gaussian ($2.69$ DU). The standard and super Gaussian retrievals show better correlation coefficient ($0.85$), followed by the stretched preflight ($0.84$). When comparing the TOC500, the relationships between different slit functions are similar to TOC.




For the mean absolute biases, standard Gaussian < stretched preflight ≈ super Gaussian < preflight. However, the correlation coefficient for the super Gaussian becomes slightly higher than the standard Gaussian.

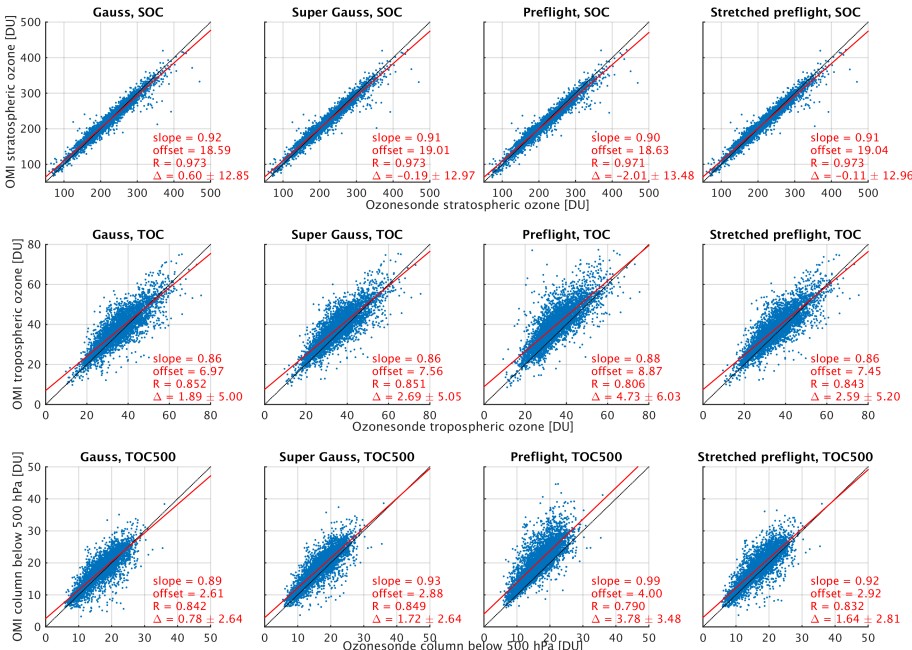

**Figure 9.** Scatter plots of OMI retrieved ozone partial columns vs. the ozonesonde data for SOC (first row), TOC (second row), and TOC500 (third row). The four columns represent retrievals using standard Gaussian, super Gaussian, the preflight, and stretched preflight slit functions. The slope and offset of the linear regression, correlation coefficient (R), and the mean bias $\pm 1\sigma$ ($\Delta$) are shown for each subplot. The linear regressions are shown as red lines, and the black dash line is the 1:1 line. The number of successful validations is 4056 (same as for all slit functions, see Fig. 4).

As shown by Fig. 9, the ozone partial columns retrieved using the preflight slit functions consistently show the lowest performance compared to the derived on-orbit slit functions. Figure 10 illustrates the vertical distributions of biases between OMI and the ozonesonde profiles. The left panel shows the median biases between OMI profiles retrieved using four different slit functions and the ozonesondes; the right panel shows the standard deviation of the differences between different retrievals and ozonesonde data. The median bias of the a priori and the a priori error are also plotted for reference. The retrieval using preflight slit functions shows significant positive bias in the stratosphere, negative bias at upper troposphere/lower stratosphere (UT/LS), and positive bias in the lower troposphere. The retrieval using preflight slit functions also has much larger variations. The median biases show different vertical distributions between standard Gaussian and super Gaussian/stretched preflight with no much altitude-dependent variation for the standard Gaussian, and almost the same larger oscillations for the latter two. Unlike the median biases, the standard deviations of biases are very similar for retrievals using these three on-orbit slit function options.





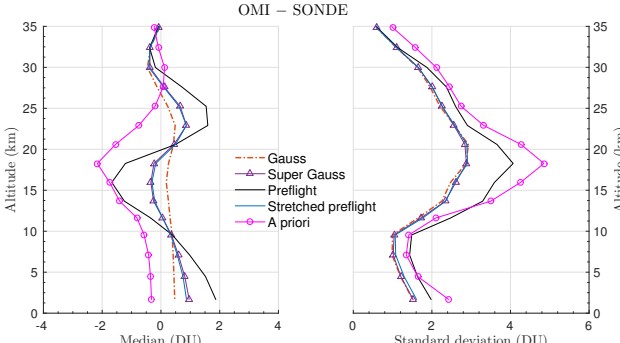

**Figure 10.** (Left) Median biases of OMI profiles retrieved using four different slit functions compared to the ozonesondes. The median difference between a priori profile and the ozonsonde data is also shown. (Right) Standard deviation of the differences between different retrievals and ozonesonde data. The a priori error is shown in the right panel.

The bias profiles between OMI and ozonesonde are further grouped according to the OMI cross-track positions, and the medians of bias profiles at each cross-track position are shown in the first row of Fig. 11. The ozone profiles retrieved using the preflight slit functions show substantial cross-track dependent biases; the bias profile shown in Fig. 10 can be largely attributed to the biases near the edges of the cross-track positions. The second and third rows of Fig. 11 present the fitting residual

root-mean-square (RMS) of UV1 and UV2 fitting windows, respectively. Although the on-orbit slit functions show the largest difference from the preflight at the UV1 band (Fig. 5), the fitting residuals at the UV1 band using different slit functions are very similar; the residual RMS values using the preflight slit functions are only marginally larger than retrievals using on-orbit slit functions. The insensitivity of fitting residuals to slit functions at the UV1 band is likely due to the smoothness of the ozone absorption cross-sections. The residual RMS at the UV2 band for the retrieval using the preflight slit functions, however, show

a distinct U-shaped cross-track distribution, very different from the other retrievals. Apparently, the preflight slit functions are not accurate near the edges of the cross-track positions in this two-band test case. It should be emphasized that in this case the UV2 radiance spectra and preflight slit functions are co-added from 60 to 30 cross-track positions. The co-adding of adjacent radiance spectra defined at different wavelength grids introduces some effective broadening of slit functions, which may partially explain the mismatch of slit functions for the preflight retrieval at UV2. Therefore, the UV2-only case is also

tested to avoid co-adding and test the UV2 slit functions only.

## 5.2 The UV2-only case

When only using the UV2 window in the ozone profile retrieval, the DFS is reduced from 6–7 to 1–2 due to the lost of most stratospheric information content from UV1. Figure 12 compares the SOC (first row) and TOC (second row) from OMI with ozonesondes. Given the low DFS, only the SOC and the TOC are validated. Similar to the two-band case, the retrieval using the

preflight slit functions shows the largest absolute biases and lowest correlation with ozonesondes for both TOC and SOC. The standard Gaussian retrieval consistently shows higher correlation coefficients and lower standard deviations of biases compared




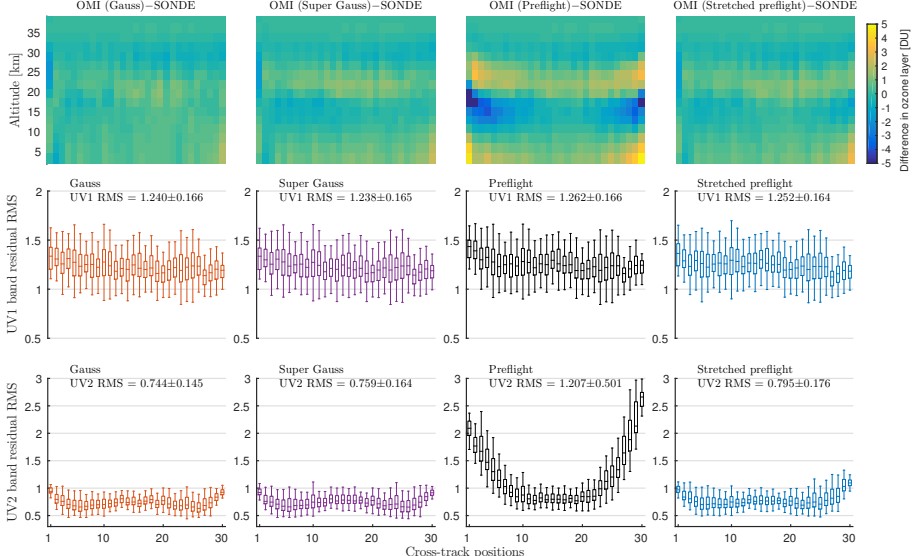

**Figure 11.** (First row) Medians of difference profiles between OMI and ozonesondes at each cross-track position. (Second row) Distributions of fitting residual RMS at each cross track position for the UV1 band. (Third row) Same as the second row but for the UV2 band. The mean and standard derivation of all residual RMS are labeled in each subplot at row 2–3.

to the other two forms of derived on-orbit slit functions that are supposed to better represent the true slit function shapes in UV2 (see Fig. 2). The only exception is that the stretched preflight retrieval shows marginally better correlation coefficient than the standard Gaussian for the TOC.

Figure 13 compares the medians and standard deviations of bias profiles from different retrievals, similar to Fig. 10. The
retrieval using the preflight slit functions also shows the largest biases and standard deviations. The three derived on-orbit slit functions show different altitude-dependency in the median biases with the bias of stretched preflight similar to super Gaussian in shape but smaller in absolute value. The standard Gaussian shows the lowest standard derivations of the bias profiles, followed by stretched preflight and super Gaussian.

Figure 14 shows the cross-track distributions of median bias profiles and fitting residual RMS for ozone-profile retrievals
using the four different slit functions, similar to Fig. 11 but for UV2 only. The preflight retrieval again stands out with significant cross-track dependent biases, larger towards the edges of the cross-track positions. The RMS of the retrieval using preflight slit functions also show U-shaped cross-track dependency, but smaller than those in Fig. 11 due to no broadening of effective slit functions from coadding. These large RMS at both edges are mitigated by fitting a stretch to the preflight slit functions. The mean RMS of using super Gaussian and stretched preflight slit functions are slightly better than using the standard Gaussian,
indicating that generally a broad-top slit function can better model the OMI spectra. However, the retrieval using standard Gaussian slit functions shows the smallest variations of biases and variations of residual RMS, which is not fully understood. It is possible that the slit function in radiance measurements cannot be fully represented by the on-orbit slit functions derived



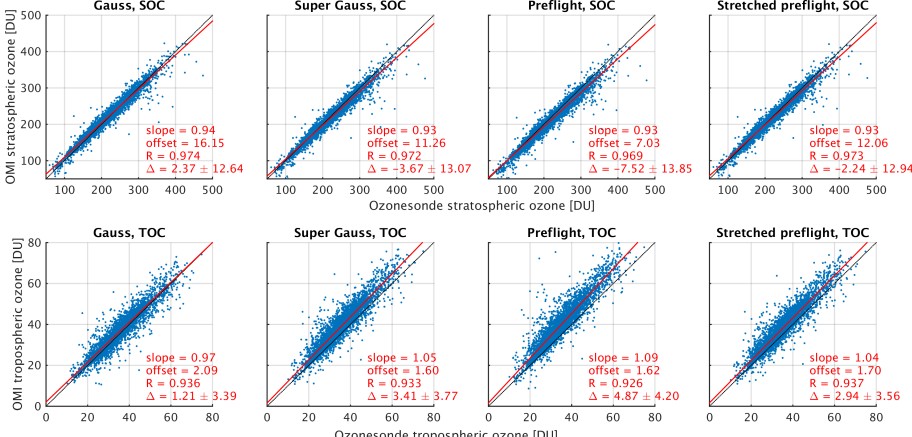

**Figure 12.** Similar to Fig. 9 but for the UV2-only case. Only SOC and TOC are shown. The number of successful validations is 4461 (same as for all slit functions, see Fig. 4).

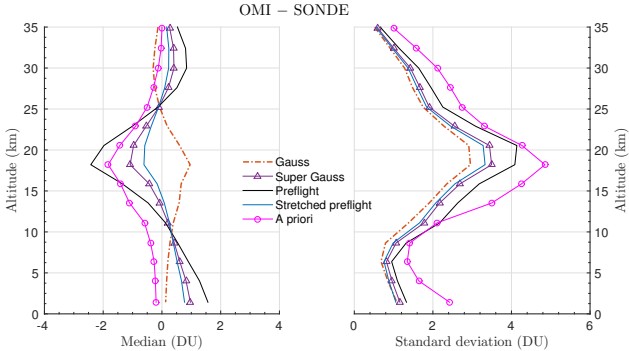

**Figure 13.** Same to Fig. 10 but for the UV2-only case.

from the solar irradiance due to scence inhomogeneity (Voors et al., 2006; Noël et al., 2012), unaccounted stray lights, or intra-orbit slit function changes (observed in GOME-2 by Beirle et al. 2017) in earthshine spectra.

# 6 Conclusions

The accurate characterization of slit functions is essential for the spectral calibration of space-borne grating spectrometers and the retrieval of the Earth's atmospheric constituents. We derive on-orbit slit functions by fitting the OMI irradiance spectra with a high-resolution solar reference spectrum and various assumptions on slit function forms, including standard and super Gaussian functions and a homogeneous stretch to the preflight slit functions. The on-orbit slit functions derived from multi-year averaged OMI solar irradiance show U-shaped cross-track dependences at the UV bands that cannot be fully represented by the preflight slit functions. The FWHMs of the stretched preflight slit functions of detector pixels at large viewing angles





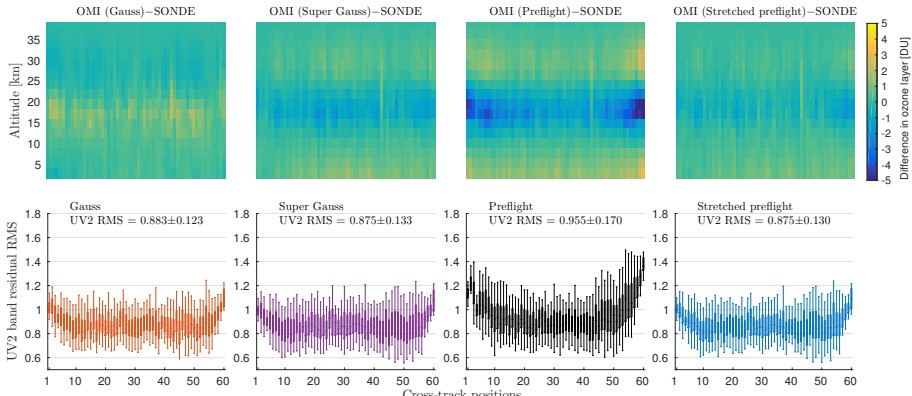

**Figure 14.** Similar to Fig.11 but for the UV2 band only. Note there are 60, instead of 30 cross-track positions.

are up to 20 % larger than the nadir ones for the UV1 band and 5 % larger for the UV2 band. When fitting super-Gaussian slit functions in the UV2 band, the cross-track variations of FWHM are much smaller, but the cross-track variations of the shape parameter $k$ are significant. The derived on-orbit slit functions using daily OMI solar irradiance from 2004 to 2016 show little temporal variations after taking account of the impact of solar activity. Overall, these temporal variations for non-RA rows,

if they exist, are no more than 1 % and much smaller than differences between the preflight and on-orbit slit functions for all three OMI bands. The RA rows also did not show any significant temporal variations before the major RA events in 2009.

Considering the insignificant temporal variations of on-orbit slit functions, the slit functions derived from the multi-year average OMI solar irradiance are applied in the SAO ozone-profile retrieval, and the results are compared with the retrieval using the preflight slit functions. Two cases are tested: one is to keep the same options of the operational algorithm whenever

possible, and the other one is to use only the UV2 band to avoid coadding the UV2 spectra and eliminate the interaction between UV1 and UV2. The retrievals using derived on-orbit slit functions consistently show smaller biases and better correlations with the ozonesonde validations in both cases. Although the on-orbit slit functions show larger difference from the preflight in the UV1 band, the impact of slit functions on the ozone-profile retrieval is dominated by the UV2 band, due to the more complicated structure of ozone absorption cross-section in UV2. The UV-2 only test case has direct implications for other OMI

products that use the UV2 band, such as $SO_2$, HCHO, and BrO. The on-orbit slit functions of the OMI VIS band also have cross-track discrepancies compared to the preflight, although less significant. Future comparisons of retrievals using different slit functions will be performed in the VIS band with well-validated algorithms, such as the water vapor retrieval (Wang et al., 2016).

It is challenging to characterize the slit functions of 2-D detectors of OMI-like instruments, as the slit functions vary in

both the wavelength (the column dimension) and cross-track viewing dimension (the row dimension) and more critically, may vary over time. Future work will involve characterizing the differences between the slit functions derived from solar irradiance and the slit functions of earthshine radiance. These differences may be caused by scene heterogeneity, differences in stray light between irradiance and radiance, and intra-orbit instrumental changes. It is possible to linearize the slit function fitting





by constructing "pseudo-absorbers" based on derivatives of slit functions and including them in the radiance fitting (Beirle et al., 2017). Accurate knowledge of the on-orbit slit functions, as demonstrated in this work, will also be important for the near-future missions that have more spatial pixels and higher retrieval targets than OMI (TROPOMI/Sentinel-5p, Sentinel-5, Sentinel-4, GEMS, and TEMPO).

5   *Acknowledgements.*  This paper is supported by NASA's Atmospheric Composition: Aura Science Team program (sponsor contract numbers NNX14AF16G and NNX14AF56G). We acknowledge Sergey Marchenko and Matthew DeLand at NASA and Science Systems and Applications, Inc. for making the solar proxy index available at https://sbuv2.gsfc.nasa.gov/solar/omi/. We thank Sergy Marchenko for discussions on the OMI solar data products. We thank Caroline Nowlan, Christopher Chan Miller, and Huiqun Wang at the Smithsonian Astrophysical Observatory (SAO), Steffen Beirle at the Max Planck Institute for Chemistry (MPI-C), and Can Li at NASA GSFC for helpful

10   discussions on slit function parameterizations. The OMI International Science Team is acknowledged for making OMI L1B data available at https://disc.gsfc.nasa.gov/Aura/data-holdings/OMI/. We also thank the ozonesonde providers and their funding agencies for making the ozonesonde measurements, as well as the Aura Validation Data Center (AVDC), World Ozone and Ultraviolet Radiation Data Center (WOUDC), and the Southern Hemisphere ADditional OZonesonde (SHADOZ) for archiving the ozonesonde data.



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
