# Peer review of "Deriving the slit functions from OMI solar observations and its implications for ozone-profile retrieval"

_Atmospheric Measurement Techniques, 2017_

## Referee Comment (RC1) · Anonymous Referee #1 · 22 Jun 2017

The article, "Deriving the slit functions from OMI solar observations and its implications for ozone-profile retrieval", by Kang Sun et al., addresses two important subjects.

1. Until recently, a proper characterization of the on-orbit evolution of the OMI's instrument transfer function (ITF) was lacking. This subject was discussed in Beirle et al. (2017) and Schenkeveld et al. (2017), however assessing the OMI ITF via different means.

2. The article also breaches the important and rarely discussed subject of the potential impact of the variable ($\sim$11yr cycle) solar spectrum on the ITF characterization.

This is a comprehensive and timely study that deserves prompt publication.

[Figure]

———————————————————————————— The final revision may benefit from addressing the following points:

1. Over the OMI mission time there are small, but systematic wavelength shifts (e.g., Schenkeveld et al. 2017) that may affect the ITF evaluations, the UV1 range in particular. The ITF fitting algorithm includes the wavelength shift/squeeze terms (p.4,l.7). How do these terms compare to the estimates from Schenkeveld et al. (2017)? Decision about the optimal choice of ITF form should rely on a multitude of criteria (see below). The temporal, spatial (FOV, i.e., row-wise) and wavelength dependence of the shifts may be one of such criteria.

2. The solar irradiances are practically row-anomaly free in the UV2 and VIS channels. The current study uses the 3-year averaged reference spectrum, yy2005-2007. Would the UV2 and VIS fitting trends change if the averaged solar-minimum spectrum (∼mid2007- ∼mid-2009) is used instead? This deserves a comment, potentially strengthening the author's conclusion that most of the detected temporal ITF variability is related to the Solar Cycle.

3. p.5., l.6: 'The OMI ozone-profile retrievals have substantially higher relative accuracy than other OMI products...'. The proposed O3-based validation clearly shows the need for adjustment of the pre- flight ITF values. The study, however, provides no clues to what parametric ITF form could be the most beneficial to the various trace-gas retrieval algorithms that may have far higher sensitivity to the ITF changes. As formulated, this extensive study falls beyond the scope of the paper. However, the revised text may include some additional stats that quantify performance of the explored ITF approximations. Besides the wavelength shifts (as mentioned above), the temporal and x-track behavior of the fitting residuals may help to decide about the optimal ITF representation. In order to provide a compact, but conscious record, such fitting-residual stats could be agglomerated to within the OMI channel, i.e., to be averaged for the UV1, UV2 and VIS ranges. They may be shown in an appendix, per author's choosing.

4. The super-Gaussian approach is rapidly 'gaining traction', considering the number of applications adopting this particular ITF form. Fig. 5 shows, however, that in the OMI case this approach leads to more noisy (far more, e.g., the 1st window in the UV2 channel, the last window of VIS) x-track behavior. Is this because the other methods are less sensitive or because the super-Gaussian parametrization is less stable? I suspect the latter. This should be verified and commented on by: (a) constructing an alternative solar reference spectrum (see point #2 above) and showing (probably, in the appendix) the similar to Fig. 5 plot; (b) comparing the temporal (e.g., 1-year blocks) behavior of the fitting parameters for all applied ITF forms and all RA-free rows/spectral regions.

5. Based on the additional metrics suggested in points #1-4, please provide, if feasible, a summary (preferably in a tabular form) of performance of different ITF approximations for all 3 OMI channels.

6. Also, please add a statement about the x-track behavior in the VIS channel to the 1st par. of Conclusions, in line with the UV1 and UV2 conclusions.

———————————————————————————————————— Minor suggestions/corrections:

Abstract, p.1,l.7: '... are up to 20 %...' should be changed to '... are up to 30 %...' (cf. Fig. 5, upper-left).

Abstract, p.1,l.8: I suggest adding '... and practically flat in VIS. Nonetheless...'

p.3,l.3: '...description of the RA can be found in...'

p.4,l.3: '... ($Lambda\_1$ and $A\_1$ are zero).' The original reads '... ($Lambda\_0$ and $A\_1$ are zero).'

p.6,l.10: '... are kept the same as in the operational algorithm...'

p.8,l.6: '... by up to 30 %...'

p.10,l.7: '...changes mostly due to variations in faculae.'

p.14,l.19: '...due to the loss of most...'

p.17,l.1: '... by up to 30 %...'

---

## Referee Comment (RC2) · R. Lang (Referee) · 10 Jul 2017

The paper by Sun et al, is an important contribution to the currently expanding field of studying the spectral response functions – or slit-functions – of satellite based hyper-spectral instruments measuring in the UV to near or short-wave infra-red region in-orbit. Until recently, and for this type of instrumentation for which a very high accuracy of slit-function characterisation is required in order to meet the accuracy requirement of the level-2 products, predominantly on-ground pre-flight slit-function data has been used. However, since both the launch, as well as instrumental changes notably in the thermal environment, may often significantly affect the spectral response of the instrument, it

is important to be able to derive the slit-function data also from in-orbit data, with a reasonable to high accuracy.

The paper by Sun et al. explores various slit-function functional forms (stretch, Gaussian, and super-Gaussian), which can be used for in-orbit characterization using a high spectral resolution solar reference spectrum together with in-flight measured solar spectra. The paper describes the main differences between in-orbit derived slit-functions and the pre-flight derived slit-function for the Ozone Monitoring Instrument (OMI) on-board the AURA platform. After a thorough discussion of the observed differences they apply the various slit-functions to their ozone profile retrieval algorithm in order to evaluate their performance by comparing the results to ground based ozone-sonde network data. The conclusion of this work is that there, are partially significant, differences in the derived slit-function parameters between on-ground and in-orbit, but less between their various functional forms. The paper shows, that OMI-retrievals can benefit, from applying in-orbit derived slit-functions, depending on the spectral region applied. The paper also confirms that the OMI instrument is sufficiently stable that a fixed set of in-orbit derived slit-functions suffices to improve the retrievals for the full OMI records.

I can recommend the paper for publication in AMT, since it is a significant contribution to improving our practical knowledge on how to derive in orbit slit-functions and by this improve the level-2 products from the increasing suite of hyper-spectral UVNS instruments. I have only a few comments, which may require minor revisions of the paper.

1) Overall there is a quite thorough description of the technical approach to derive in-orbit slit-functions. However, the paper lacks some detail on the minimization procedure for the slit-function fitting. What kind of minimization has been applied, what kind of site constrains have been applied, and also with respect to the spectral band edges. The paper could provide some more details on these aspects in section 2.1.1.
2) The functional form of the slit-function used for pre-flight measurements and its performance with respect to the Gaussian and the super-Gaussian, which are used for the in-orbit measurements, is not discussed in the paper. In principle one should test the three functional forms on the same data-set, e.g. the pre-flight data-set, in order to see if they provide the same results, i.e. if the changes observed between on-ground and in-orbit are clearly instrument related, or due to the usage of different functional forms. In addition, in order to avoid that the worse performance, e.g. for a spectrally changing asymmetry, is just a result of missing information (i.e. due to the null space of the problem using the solar spectrum, which itself is spectrally dependent) it could be considered to derive spectrally dependent parameter from fitting the on-ground measured, dedicated slit-function spectra, with higher information content for the spectral response.

3) I would propose to move the section on the comparison of the results for the pre-flight and in-flight slit-function parameters (section 3 and 4) directly after the description of the method (section 2.1), since for me these sections form a closed entity of instrument characterization including its result, which is then applied, in a next step, for an evaluation of its respective performance in the level-2 retrievals.

4) In section 4 on the temporal evolution of the in-orbit slit function, in case there would be a small in-orbit seasonal variation of the instrument slit-function, e.g. due to temperature variations, how would this affect the MgII index itself. Should we then not potentially expect a correlation of the results in first place, but not due to the solar variability but due to the slit-function changes itself? 5) At the end of section 4, the authors speculate about the reason why "the retrieval using standard Gaussian slit functions shows the smallest variations of biases and variations of residual RMS, which is not fully understood." And they hint at issues like stray-light, scene in-homogeneity or intra-orbit changes. But why would this not affect the super-Gaussian and the stretched pre-flight functional forms in the same way?

Editorial:

Page 14, line 17, "lost" -> loss

---

## Author Comment (AC1) · 24 Aug 2017

Response to Referee #1:

We appreciate the very helpful feedback from the referee. The referee's comments are listed in *italics*, followed by our response in blue. New/modified text in the manuscript is in **bold**. The manuscript has been re-organized following referee #2's suggestion. Please note that some figure numbers and line numbers have been changed from the original manuscript.

*1. Over the OMI mission time there are small, but systematic wavelength shifts (e.g., Schenkeveld et al. 2017) that may affect the ITF evaluations, the UV1 range in particular. The ITF fitting algorithm includes the wavelength shift/squeeze terms (p.4,l.7). How do these terms compare to the estimates from Schenkeveld et al. (2017)? Decision about the optimal choice of ITF form should rely on a multitude of criteria (see below). The temporal, spatial (FOV, i.e., row-wise) and wavelength dependence of the shifts may be one of such criteria.*

The wavelength shift terms fitted from the irradiance (figure below) are very similar to the wavelength shift derived from the radiance channel shown in Fig. 33 in Schenkeveld et al. (2017). The wavelength shift terms derived from multiple symmetric slit function fits show very similar trends. The wavelength shift is better constrained in the UV1 irradiance channel than UV1 radiance channel because there is no ozone absorption in the irradiance. To clarify, the following sentence was added (Page 4, Line 7 of the original manuscript):

"**The wavelength shift terms derived here are consistent with the spectral calibration trends using the OMI radiance (Fig. 33 in Schenkeveld et al. 2017). No significantly different trends of spectral shifts are observed for different symmetric slit function fits.**"

[Figure]

**Figure 1. Wavelength shifts derived by fitting OMI UV1, UV2, and VIS irradiance spectra using super Gaussian slit functions. The temporal trends of wavelength shifts derived from standard Gaussian and stretched preflight are very similar to this plot.**

*2. The solar irradiances are practically row-anomaly free in the UV2 and VIS channels. The current study uses the 3-year averaged reference spectrum, yy2005-2007. Would the UV2 and VIS fitting trends change if the averaged solar-minimum spectrum(~mid2007-~mid-2009) is used instead? This deserves a comment, potentially strengthening the author's conclusion that most of the detected temporal ITF variability is related to the Solar Cycle.*

We have tested the fitting using averaged OMI solar irradiance spectra from 10/2004 to 6/2007 (RA free, moderate solar activity), from 1/2007 to 1/2009 (insignificant RA, solar minimum), and from 1/2010 to 9/2012 (significant RA, significant solar activity). The results of 2004-mid 2007 are essentially the same as Fig. 5 in the original manuscript (we used a very similar period, 2005-2007, in Fig. 5 of the original manuscript). We updated Fig. 5 in the original manuscript (Fig. 3 in the revised manuscript) using the new results. In previous Fig. 5, we used the first principal component of multiple year solar spectra for the VIS band, but multi-year average for UV1 and UV2. In the revision, we update the VIS irradiance with multi-year averaging, to be consistent with UV1 and UV2. We have also included the super Gaussian fitting for the UV1 band. The results for 2007-2009 and 2010-2012 are very similar and have been included as Appendix A in the revised manuscript.

The following sentences have been added in page 8, line 20 of the original manuscript:

"**We have also included the fitting results using average OMI solar spectra from 1/2007 to 1/2009 (insignificant RA, solar minimum), and from 1/2010 to 9/2012 (significant RA, significant solar activity) in the Appendix A. The cross-track features are very similar during these periods, indicating that the cross-track dependent features observed on-orbit are not due to any temporal effects (e.g., RA, solar activities**)."

The updated Fig. 5 of the original manuscript (Fig. 3 of the revised manuscript):

[Figure]

**Figure 2. Updated Fig. 5 in the original manuscript (Fig. 3 in the revised manuscript). It shows the cross-track fitting results using average solar spectra from 10/2004 to 6/2007.**

*3. p.5., l.6: 'The OMI ozone-profile retrievals have substantially higher relative accuracy than other OMI products...'. The proposed O3-based validation clearly shows the need for adjustment of the pre- flight ITF values. The study, however, provides no clues to what parametric ITF form could be the most beneficial to the various trace-gas retrieval algorithms that may have far higher sensitivity to the ITF changes. As formulated, this extensive study falls beyond the scope of the paper. However, the revised text may include some additional stats that quantify performance of the explored ITF approximations. Besides the wavelength shifts (as mentioned above), the temporal and x-track behavior of the fitting residuals may help to decide about the optimal ITF representation. In order to provide a compact, but conscious record, such fitting-residual stats could be agglomerated to within the OMI channel, i.e., to be averaged for the UV1, UV2 and VIS ranges. They may be shown in an appendix, per author's choosing.*

Following the reviewer's suggestion, we added the following figure to the Appendix (Fig. A.3 of the revised manuscript) to show the fitting residuals using different slit functions at different cross-track positions and early/late in the mission. For fitting residuals of UV2 and VIS bands (relevant for trace gases), super Gaussian ≈ stretched preflight < standard Gaussian. For the UV1 band, super Gaussian shows smaller fitting residual than standard Gaussian (stretched preflight is

identical to standard Gaussian for UV1), but unstable cross-track features in the fitting results, possibly due to the correlation between width and shape parameters when the shape parameter is small.

[Figure]

**Figure 3. The residual RMS when fitting different slit functions over the OMI UV1, UV2, and VIS bands. The lines without circles are fitting results using solar spectra averaged over 2005; the lines with circles are fitting results using solar spectra averaged over 2015.**

*4. The super-Gaussian approach is rapidly 'gaining traction', considering the number of applications adopting this particular ITF form. Fig. 5 shows, however, that in the OMI case this approach leads to more noisy (far more, e.g., the 1st window in the UV2 channel, the last window of VIS) x-track behavior. Is this because the other methods are less sensitive or because the super-Gaussian parametrization is less stable? I suspect the latter. This should be verified and commented on by: (a) constructing an alternative solar reference spectrum (see point #2 above) and showing (probably, in the appendix) the similar to Fig. 5 plot; (b) comparing the temporal (e.g., 1-year blocks) behavior of the fitting parameters for all applied ITF forms and all RA-free rows/spectral regions.*

The super Gaussian indeed yields unstable cross-track features, mainly in the UV1 band and window 1 of the UV2 band. We update the following sentences in page 8, line 3 of the original manuscript:

"**For the UV1 band (the first row of Fig. 3), the preflight slit functions are standard Gaussian, so fitting a stretch to the preflight slit functions is identical to fitting a standard Gaussian. The super Gaussian shows unstable cross-track features, favoring the use of standard Gaussian in the UV1 band.**"

Although the super Gaussian results look unstable in the cross-track dimension in some windows, they are generally stable over time. The following sentences have been added in page 8, line 20 of the original manuscript:

**"For the UV1 band and window 1 of the UV2 band, the super Gaussian results show unstable cross-track features, likely due to the correlation between the width and shape parameters, but appear to be generally invariant over time."**

(a) Two alternative solar reference spectra have been constructed and shown in Appendix A. The two plots in Appendix A are also shown below.

[Figure]

Figure 4. Cross-track fitting results using average solar spectra from 1/2007 to 1/2009. This is Fig. A.1 in the revised manuscript.

[Figure]

**Figure 5. Cross-track fitting results using average solar spectra from 1/2010 to 9/2012. This is Fig. A.2 in the revised manuscript.**

(b) The temporal behavior in ~2 year block can be found by comparing Figs. 2, 4, and 5 of this document (same as Fig. 3, Fig. A.1, and Fig. A.2 of the revised manuscript).

*5. Based on the additional metrics suggested in points #1-4, please provide, if feasible, a summary (preferably in a tabular form) of performance of different ITF approximations for all 3 OMI channels.*

The following summary was included at the end of Section 3:

"**We have also included the fitting results using average OMI solar spectra from 1/2007 to 1/2009 (insignificant RA, solar minimum), and from 1/2010 to 9/2012 (significant RA, significant solar activity) in Fig. A.1 and Fig. A.2 in the Appendix. The cross-track features are very similar during these periods, indicating that these cross-track dependent features observed on-orbit are not due to any temporal effects (e.g., RA, solar activities). For the UV1 band and window 1 of the UV2 band, the super Gaussian results show unstable cross-track features, but appear to be generally invariant over time. Figure A.3 further compares the fitting residuals agglomerated to within each OMI band using different slit functions at different cross-track positions and early/late in the mission. The super Gaussian and**

**stretched preflight slit functions show smaller fitting residuals than standard Gaussian for the UV2 and VIS bands, although an ozone retrieval test using these three slit function forms at window 1 of UV2 band shows only small differences (see Sect. 5. 3).**"

*6. Also, please add a statement about the x-track behavior in the VIS channel to the 1st par. of Conclusions, in line with the UV1 and UV2 conclusions.*

The following has been added to page 17, line 3 of the original manuscript:

"**No significant discrepancy was found in the VIS band except for the first fitting window (380--402 nm), where a moderate reverse-U-shaped cross-track dependency of derived slit function width was present.**"

*Abstract, p.1,l.7: '... are up to 20 %...' should be changed to '... are up to 30 %...' (cf. Fig. 5, upper-left).*

Revised.

*Abstract, p.1,l.8: I suggest adding '... and practically flat in VIS. Nonetheless...'*

Revised.

*p.3,l.3: '...description of the RA can be found in...'*

Revised.

*p.4,l.3: '... (Lambda_1 and A_1 are zero).' The original reads '... (Lambda_0 and A_1 are zero).'*

The original is actually correct. $A_1 = 0$ vanishes the second term on the right hand side of the equation, so the value of $\lambda_1$ is not useful anymore. We then need $\lambda_0 = 0$ so that the left standard Gaussian term centers at zero.

*p.6,l.10: '... are kept the same as in the operational algorithm...'*

Revised.

*p.8,l.6: '... by up to 30 %...'*

Done.

*p.10,l.7: '...changes mostly due to variations in faculae.*

Revised.

*p.14,l.19: '...due to the loss of most...'*

Revised.

*p.17,l.1: '... by up to 30 %...'*

Done.

---

## Author Comment (AC2) · 24 Aug 2017

Response to Referee #2:

We appreciate the very helpful feedback from the referee. The referee's comments are listed in *italics*, followed by our response in blue. New/modified text in the manuscript is in **bold**. The manuscript has been re-organized following the referee's suggestion. Please note that some figure numbers and line numbers have been changed from the original manuscript.

*1) Overall there is a quite thorough description of the technical approach to derive in orbit slit-functions. However, the paper lacks some detail on the minimization procedure for the slit-function fitting. What kind of minimization has been applied, what kind of site constrains have been applied, and also with respect to the spectral band edges. The paper could provide some more details on these aspects in section 2.1.1.*

The following details have been provided in section 2.1.1. (page 4, line 7 of the original manuscript):

"**The fitting applies a weighted Levenberg-Marquardt nonlinear least square algorithm to minimize the sum of squares of fitting residuals weighted by OMI spectral uncertainties. The high-resolution reference spectrum is extended by 5 nm beyond the fitting window edges to mitigate the edge effect.**"

*2) The functional form of the slit-function used for pre-flight measurements and its performance with respect to the Gaussian and the super-Gaussian, which are used for the in-orbit measurements, is not discussed in the paper. In principle one should test the three functional forms on the same data-set, e.g. the pre-flight data-set, in order to see if they provide the same results, i.e. if the changes observed between on-ground and in-orbit are clearly instrument related, or due to the usage of different functional forms. In addition, in order to avoid that the worse performance, e.g. for a spectrally changing asymmetry, is just a result of missing information (i.e. due to the null space of the problem using the solar spectrum, which itself is spectrally dependent) it could be considered to derive spectrally dependent parameter from fitting the on-ground measured, dedicated slit-function spectra, with higher information content for the spectral response.*

We have indeed tested the functional form used for pre-flight measurements (i.e., Eq. 1 of the manuscript). The results show consistent cross-track features in the UV1 and UV2 bands, similar to the other functional forms (standard Gaussian, super Gaussian, and hybrid Gaussian as in Nowlan et al. 2016). However, the results of the preflight functional form and the hybrid Gaussian (both are hybrid forms of a standard Gaussian and a super Gaussian with $k = 4$) are unstable due to fitting too many parameters and correlation among some parameters. This issue became more significant later in the OMI mission, when the solar spectra SNR degraded. In addition, the asymmetry terms (either relative offsets between standard and super Gaussian or the uneven widths) compete with the spectral shift term and make the spectral calibration much noisier than the symmetric functional forms. It has been shown that the OMI slit functions are effectively symmetric (Beirle et al. 2017). As such, we did not use the results of the preflight functional form. To clarify this point, we updated the following sentences at page 4, line 12 of the original manuscript:

"**These function forms, as well as the functional form used to parameterize the OMI preflight slit functions (Eq. 1), were also tested for OMI. The (a)symmetric standard Gaussian, super Gaussian, and fitting a homogeneous stretch to the preflight slit functions were found to produce stable fitting results. The other function forms (stretch/sharpen, hybrid Gaussians, and the functional form of Eq. 1) are unstable due to fitting too many parameters and the correlation of some parameters. The stability issue became more significant later in the OMI mission, when the signal-to-noise ratio (SNR) of solar spectra was degraded.**"

The changes between on-ground and in-orbit slit functions are most significant in the cross-track dimension. The on-ground slit functions show little cross-track dependency, whereas the in-orbit slit functions show U-shaped cross-track dependency (Fig. 2 and Fig. 5 of the original manuscript). This difference is independent of the functional forms we choose to fit the in-orbit slit functions. To clarify this point, we added the following sentence to page 8, line 16 of the original manuscript:

"**The cross-track dependency is significant for all functional forms used in the fitting, indicating that the changes observed between on-ground and in-orbit is instrument related, not due to usage of different fitting functional forms.**"

We did not observe any spectrally changing asymmetry when fitting both symmetric and asymmetric slit functions in a range of spectral windows (Fig. 5 of the original manuscript). The current preflight slit functions were derived from the on-ground measured, dedicated slit-function spectra, and we have shown that the preflight slit functions do not capture the U-shaped cross-track dependency of in-orbit slit function. Therefore we do not think it is necessary to re-derive the preflight slit functions based on on-ground measured, dedicated slit-function spectra.

*3) I would propose to move the section on the comparison of the results for the preflight and in-flight slit-function parameters (section 3 and 4) directly after the description of the method (section 2.1), since for me these sections form a closed entity of instrument characterization including its result, which is then applied, in a next step, for an evaluation of its respective performance in the level-2 retrievals.*

The manuscript has been reorganized following the referee's suggestion. The original Subsection 2.2: "SAO OMI ozone-profile retrieval algorithm and validation" was moved to Section 5.

*4) In section 4 on the temporal evolution of the in-orbit slit function, in case there would be a small in-orbit seasonal variation of the instrument slit-function, e.g. due to temperature variations, how would this affect the MgII index itself. Should we then not potentially expect a correlation of the results in first place, but not due to the solar variability but due to the slit-function changes itself?*

The MgII index has been thoroughly studied and compared with solar indices derived from other satellite instruments in DeLand and Marchenko (2013). No seasonal trends can be found in the OMI MgII index. The following plot shows the time series of different OMI solar indices. Only the 11-year and 27-day solar cycles are significant.

[Figure]

**Figure 1. Time series of OMI solar indices. Data are available at https://sbuv2.gsfc.nasa.gov/solar/omi/.**

To clarify, the following sentence was added to page 10, line 7 of the original manuscript:

"**No seasonal variations due to instrument temperature effects can be found in the MgII index. Given the complex nature of solar activity, it is highly unlikely that any potential factors that may contribute to slit function change (e.g., instrument degradation, RA effects, and instrument temperature variations) are correlated with the Mg II index.**"

*5) At the end of section 4, the authors speculate about the reason why "the retrieval using standard Gaussian slit functions shows the smallest variations of biases and variations of residual RMS, which is not fully understood." And they hint at issues like stray-light, scene inhomogeneity or intra-orbit changes. But why would this not affect the super-Gaussian and the stretched pre-flight functional forms in the same way?*

It should be at the end of section 5. The slit functions are first derived from solar irradiance, and then applied to the ozone profile retrieval. The fact that the super Gaussian and stretched preflight slit functions fit the solar spectra better than standard Gaussian, but show mixed results compared to standard Gaussian in the retrieval indicates that the slit functions may not be strictly identical between irradiance and radiance, due to some physics that are not completely accounted for. We think stray-light, scene heterogeneity, and intra-orbit changes may contribute. To demonstrate that super Gaussian and stretched preflight fits the UV-2 solar irradiance better than standard Gaussian, we have included a comparison of the solar irradiance fitting residuals as an appendix of the revised manuscript with the following figure:

[Figure]

**Figure 2. The residual RMS when fitting different slit functions over the OMI UV1, UV2, and VIS bands. The lines without circles are fitting results using solar spectra averaged over 2005; the lines with circles are fitting results using solar spectra averaged over 2015.**

*Page 14, line 17, "lost" -> loss*

Revised.

**References**

Beirle, S., Lampel, J., Lerot, C., Sihler, H., and Wagner, T.: Parameterizing the instrumental spectral response function and its changes by a super-Gaussian and its derivatives, Atmospheric Measurement Techniques, 10, 581–598, doi:10.5194/amt-10-581-2017, http://www. atmos-meas-tech.net/10/581/2017/, 2017.

DeLand, M. and Marchenko, S.: The solar chromospheric Ca and Mg indices from Aura OMI, Journal of Geophysical Research: Atmospheres, 118, 3415–3423, doi:10.1002/jgrd.50310, http://dx.doi.org/10.1002/jgrd.50310, 2013.

Nowlan, C. R., Liu, X., Leitch, J. W., Chance, K., González Abad, G., Liu, C., Zoogman, P., Cole, J., Delker, T., Good, W., Murcray, F., Ruppert, L., Soo, D., Follette-Cook, M. B., Janz, S. J., Kowalewski, M. G., Loughner, C. P., Pickering, K. E., Herman, J. R., Beaver, M. R., Long, R. W., Szykman, J. J., Judd, L. M., Kelley, P., Luke, W. T., Ren, X., and Al-Saadi, J. A.: Nitrogen dioxide observations from the Geostationary Trace gas and Aerosol Sensor Optimization (GeoTASO) airborne instrument: Retrieval algorithm and measurements during DISCOVER-AQ Texas 2013, Atmospheric Measurement Techniques, 9, 2647–2668, doi:10.5194/amt-9-2647-2016, http://www.atmos-meas-tech.net/9/2647/2016/, 2016.